# Mechanisms of Trichomes and Terpene Compounds in Indigenous and Commercial Thai Rice Varieties against Brown Planthopper

**DOI:** 10.3390/insects13050427

**Published:** 2022-05-01

**Authors:** Phawini Khetnon, Kanungnid Busarakam, Wissarut Sukhaket, Cholticha Niwaspragrit, Wintai Kamolsukyeunyong, Naoto Kamata, Sunisa Sanguansub

**Affiliations:** 1Department of Entomology, Faculty of Agriculture at Kamphaeng Saen, Kasetsart University, Kamphaeng Saen Campus, Nakhon Pathom 73140, Thailand; phawini_k@outlook.co.th (P.K.); kamatan@uf.a.u-tokyo.ac.jp (N.K.); 2Expert Centre of Innovative Agriculture, TISTR Technopolis, Khlong Ha, Khlong Luang, Pathum Thani 12120, Thailand; wissarut@tistr.or.th (W.S.); cholticha@tistr.or.th (C.N.); 3Biodiversity Research Centre, TISTR Technopolis, Khlong Ha, Khlong Luang, Pathum Thani 12120, Thailand; kanungnid@tistr.or.th; 4School of Environmental Sciences, University of Guelph, Guelph, ON N1G 2W1, Canada; 5National Center for Genetic Engineering and Biotechnology (BIOTEC), 113 Thailand Science Park, Phahonyothin Road, Khlong Nueng, Khlong Luang, Pathum Thani 12120, Thailand; wtkmsyy@gmail.com; 6The University of Tokyo Chiba Forest, Graduate School of Agriculture and Life Sciences, The University of Tokyo, Kamogawa, Chiba 299-5503, Japan

**Keywords:** brown planthopper, rice, trichome, volatile organic compound

## Abstract

**Simple Summary:**

Herbivorous insects and their host plants have a long history of co-evolution. Plants produce specialized morphological structures known as trichomes, which may be involved in the antibiosis and antixenosis traits of the host plant when interacting with insect herbivores. The host plant also produces chemicals that may act as a repellent or antifeedant to reduce infestation by herbivores. Several studies over the last few decades have revealed that trichomes perform an important role in plants’ defense against herbivores. However, little information is available on the physical and chemical defense of indigenous and commercial Thai rice varieties against major pests such as the brown planthopper (BPH). In this study, we found a negative relationship between the density of prickle trichomes and the BPH infestation level. The volatile organic compound (VOC) emission profiles from rice plants indicated that β-Sesquiphellandrene induced by BPH possibly repelled BPH.

**Abstract:**

Plant trichomes generally act as a physical defense against herbivore attacks and are present in a variety of plants, including rice plants. This research examined the physical and chemical defenses of rice plants against the brown planthopper (BPH), *Nilaparvata lugens* (Stål) (Hemiptera: Delphacidae). A total of 10 rice varieties were used in this study. An electron microscope was used to observe trichomes. Constitutive and induced volatile compound profiles were assessed using GC-MS analyses. The preference of BPH for volatiles from the 10 rice plants was tested using a two-choice arena olfactometer system. The density of prickle trichomes had a negative relationship with the BPH injury level. Without BPH infestation, the volatile of the most resistant rice variety (Rathu Heenati (RH)) was preferred by BPH than those of the other varieties, with the exception of Gled Plah Chawn. However, the relative BPH preference for volatiles from the RH variety decreased during BPH infestation. When rice plants were infested by BPH, the numbers of VOCs and these quantities decreased. In the RH variety, the emission of essentities found without BPH infestation ceased during infestation by BPH. During the BPH infestation, rice plants started to emit new VOCs that were not detected before the BPH infestation started. In conclusion, we discovered that rice plants defended against BPH by changing VOC components during BPH infestation and β-Sesquiphellandrene was likely the most effective component.

## 1. Introduction

Rice is a staple food throughout the world and the most important economic crop in Thailand. For over a century, rice crops in Thailand have faced destruction by the brown planthopper (BPH) *Nilaparvata lugens* (Stål) (Hemiptera: Delphacidae), which causes a large amount of leaf burning. BPHs not only attack plants physically with their piercing-sucking mouthparts, but they also transmit both grassy stunt and ragged stunt viruses that result in undesirable yields [1,2,3]. Several insecticides have been widely applied to control the BPH outbreak. This practice may initiate chemical resistance in the BPH when inappropriately used. In addition, some insecticides affect the rice crop ecosystem and the environment [4,5,6,7,8]. The adoption of resistant varieties, particularly those with robust self-defense mechanisms, is one of the most environmentally beneficial strategies of Integrated Pest Management (IPM) against BPH.

In general, plants defend themselves against herbivores by sheltering some parts with wax films, spines, trichomes, and hardened leaves, or chemically by containing or emitting organic compounds such as terpenoids and alkaloids, anthocyanins, phenols, and quinones [9,10,11].

Trichomes, the epidermal outgrowths that cover most aerial parts of plant tissues, are one of the most intriguing plant defense systems since they perform a critical role in physical and chemical mechanisms. Trichomes are classified into two types: non–glandular trichomes and glandular trichomes. Non–glandular trichomes respond to abiotic stresses by preventing dehydration and UVA–UVB damage [12], acting as a temperature stabilizer [13,14,15] and acting as a barrier against herbivores [16,17,18,19]. Glandular trichomes are composed of many cells that emit volatile substances into the environment and substances that remain on the plants’ surface to respond to both biotic and abiotic stressors through tactility and wounds. The emission or expression of volatile organic compounds (VOCs) such as terpenoids, phenylpropanoids, flavonoids, methyl ketones, and terpene mixture are released at the cuticle through glandular trichomes [20,21,22]. These VOCs are secondary metabolites chemically mediated for plant defense against insects, pest repellence [23], natural enemy attraction, and antixenosis processes of plant self–defense [24]. Terpenoid compounds of the Rathu Heenati (RH) variety and their isogenic lines were identified by extraction at 75 °C for 20 min before being subjected to GC-MS in a previous study [25]. However, those VOCs were not spontaneous emissions from plants.

Previous research on plant resistance based on physical defenses found that trichomes on cabbage, soya pods, and wheat act as physical barriers against insect herbivores [26,27,28]. There have been studies on the chemical defense of glandular trichomes in wild tomatoes, which found the release of 7-epi-zingiberene, a particular sesquiterpene that responded as a whitefly repellent [29]. However, there are relatively few instances of rice plant trichomes acting as both physical and, more importantly, chemical resistance against BPH damage.

The purpose of this study was to explore the mechanism of resistance generated by rice trichomes. To clarify this question, 10 varieties of rice were chosen which included wild varieties, commercial varieties, and a specific donor from a rice breeding program in Thailand. The resistance to injury of these varieties was evaluated by exposure to BPH infestation. The physical defense by trichomes was determined from the relationship of the BPH injury level with the length and density of trichomes. The relative preference of BPH was determined by two-choice tests between the RH variety and others. The relative preference and VOC profiles of each variety were compared between rice plants with and without BPH infestation. Finally, the physical and chemical defenses of the 10 varieties of rice plants in relation to trichomes were discussed.

## 2. Materials and Methods

### 2.1. Experimental Design

The study was carried out between 2020 and 2021 at the Lamtakhong Research Station, Thailand Institute of Scientific and Technological Research (TISTR), Pak Chong District, Nakhon Ratchasima, Thailand (14°46′23.97″ N; 101°31′10.86″ E). The overall research flow is as follows: A total of 10 varieties of rice plants were used in this study. Potted rice plants were exposed to BPH to evaluate resistance to BPH. Regarding physical defense, the length and density of trichomes were determined through microscopic observation. The relationship between the injury level and each measurement of trichomes was analyzed. Regarding chemical defense, the relative preference of BPH was evaluated by a two-choice tests between the RH variety and the others. VOCs from rice plants with and without BPH infestation were identified and quantified using GC–MS. The relative preference and the VOCs were compared before and after the exposure to BPH.

### 2.2. Materials

#### 2.2.1. Rice Plant

A total of 10 varieties of rice plants were used in this study, which included six varieties indigenous to Thailand and four other varieties (Table 1). Resistance to BPH was reported for the latter four but not for the six indigenous varieties. To compare trichomes between resistant and susceptible varieties, indigenous rice varieties were selected by referencing information on the existence of trichomes registered in a germplasm database. Seedlings of the 10 rice varieties were grown at open-air temperature with 50% relative humidity and a photoperiod of 12:12 h light/dark and used in the further experiments.

#### 2.2.2. Brown Planthopper Mass Rearing

Approximately 2000 BPHs were collected from an organic rice paddy field in Ban Kruat District, Buriram Province, northeastern Thailand, and reared at the Rice Science Center, Kasetsart University, Kamphaeng Saen Campus, Nakhon Pathom, Thailand.

The BPHs were mass-reared for propagation in an aluminum cage (width 35 cm; length 35 cm; height 35 cm) with nylon fabric and fed with the seedling stage of the TN1 variety at a temperature of 25–30 °C and 51–64% relative humidity. The second and third generations of BPHs were used for further experiments.

### 2.3. Evaluation for Brown Planthopper Resistance

The Standard Seed-box Screening (SSBS) method by IRRI [33,34] was conducted by Randomized Complete Block Design (RCBD) with six replications. At 7 days old, seedlings were exposed to 15 s or third instars BHP nymphs per seedling. The hopper burn symptom started 7 days after infestation (DAI). After that, the injury levels were scored every other day based on the Standard Evaluation System for Rice (SES) until the seedlings completely died [33,34]. The area under curve (AUC) was calculated according to the trapezoid rule [35,36] using the three SES scores at 7, 9, and 11 DAI. The percentages of AUCs of individual seedlings to the maximum AUC value (17, obtained for one individual of the variety TN1) were averaged to each variety, which was used to categorize the resistance of each variety to BPH according to the following criteria: 0–20%, highly resistant (HR); 20–40%, resistant (R); 40–60%, intermediate (I); 60–80%, susceptible (S); 80–100%, and highly susceptible (HS).

### 2.4. Morphology Observation Using Scanning Electron Microscopy (SEM)

Trichomes on the fresh leaf blade and leaf sheath at the tillering stage of selected varieties were observed using SEM (JEOL; model JCM-6000 Plus) with three replicates. They were maintained at a temperature of 4 °C to avoid evaporation and retain their freshness. Samples were attached to the sample stage with sticky carbon tape and then sputtered with gold for 30 s. The observation area was limited to 40,000 square micrometers at 500× magnification [37,38]. The trichomes were classified into five types: glandular trichomes and four types of non-glandular trichomes (prickle, macro, micro, and papillae trichomes). The length of each trichome and trichome type identified by the outer morphology were recorded. The density per square millimeter and length of each type of trichomes were determined for each of the three individual plants of each variety. The correlation between these variables and the resistance levels was afterward determined.

### 2.5. VOCs Collection and Identification of Rice Terpenes and Derivatives by the Constitutive Defense and Inducible Defense

#### 2.5.1. VOC Collections for Constitutive and Induced Defense

To collect the VOCs for constitutive defense investigation, three rice plants in a pot at the tillering stage without infestation by BPH were placed into the guillotine covered with a glass chamber to keep the pot outside the volatile collecting trap system. A volatile collecting trap (VCT) containing the absorbent was attached to the system to trap the blend of volatile compounds emitted from the rice plants. The airflow was set at 15 kilopascals of olfactometer pressure and −15 kilopascals of olfactometer vacuum. The VCTs were set up to trap volatile compounds for 3, 6, and 9 h from 9:00 AM in the preliminary test. There were no significant changes in compounds trapped after 3 h; therefore, only the results for 3 h were used in this study (Appendix A).

VOCs for induced defense were collected from rice plants under infestation by second or third instar BPHs by the same method.

#### 2.5.2. Extraction and GC-MS Analysis

The VOCs from VCTs from the 2.5.1 were extracted from the absorbent using 4 mL of dichloromethane (CH_2_Cl_2_) from the absorbent. The 300 µL of sample solution was then stored in a headspace vial and crimped for further analysis by GC-MS (Perkin Elmer, model Clarus 680 column) with Column Elite-5MS (30 m × 0.25 μm i.d. × 0.25 µm film thickness, Perkin Elmer). The oven temperature was initially set at 60 °C, then increased at a rate of 7 °C/min to a final temperature of 250 °C. Purified helium was used as the carrier gas at a 1 mL/min flow rate. For the MS analyzer, EI mass spectra were collected at 70 eV ionization voltages over the range of 45–500 m/z. The electron multiplier voltage was 70 eV. The ion source and quadrupole temperatures were set at 230 °C and 150 °C, respectively. The identification of volatile components was performed by comparing the mass spectra in the NIST MS Search Version 2.2 database [25,39].

### 2.6. Relative Preference of Brown Planthopper between RH and the Other Varieties of Rice Plants

The behavioral assays were conducted in a two-choice arena olfactometer (30 × 30 cm) connected with an air delivery system that blew the air at 1.5 L/min constant flow to determine the relative preference of BPH toward VOC blends of different varieties of rice plants. The experiment was conducted in a laboratory under fluorescent light at 25 °C. The standard resistant variety RH was compared against the other nine varieties. Potted rice plants at the tillering stage were used for this experiment. A total of 40 BPHs at second to third instars were released at the center of the two–choice arena olfactometer for 3 h. The number of BPH in the insect isolation trap (IIT) was counted as their choice of preference. The experiments were conducted in three replications by switching the direction of inlet air connected to different rice sources. The BPHs were replaced with new ones after every trial [40,41].

The response to constitutive defense was evaluated using plants without BPH infestation. On the other hand, the inlet rice plants infested by 30 BPHs at second to third instars were used to evaluate induced defense (Appendix A).

### 2.7. Statistical Analysis

All the statistical analyses were conducted using R version 4.1.2 [42]. Non–parametric statistics were employed for injury levels in the SSBS test, which was evaluated by scores. The Kruskal–Wallis ranksum test was used to test the difference in the injury scores among varieties, followed by Dunn’s multiple comparison, in which *p*-values were adjusted with the Benjamini–Hochberg method. The package ‘FSA’ [43] was used for Dunn’s Kruskal–Wallis multiple comparison. The relationship between the injury level and variables of trichomes was tested through Spearman’s rank correlation. Logistic regression was used to determine the relative preference of BPH to each variety against the RH variety. The results using RH as an opponent were included in the base model. The coefficient of each variety in the model was used to diagnose the relative preference. Analysis of variance (ANOVA) was used to identify the extent of variation in each trichome type of density and length among the rice varieties. The differences in each measurement between each combination of two varieties were tested by a multiple comparison using Duncan’s multiple range test (DMRT). The package ‘agricolae’ was used for DMRT [44].

## 3. Results

### 3.1. Screening for Brown Planthopper Resistance

According to the criteria, the result of injury levels observed from the SSBS test was categorized into four levels (Table 2). The only highly resistant (HR) and resistant (R) varieties were RH (relative AUC; 18.6%) and Suphan Buri 1 (ditto; 31.4%), respectively. Five varieties were categorized as susceptible (S) with significant differences in the AUC from the resistant and highly resistant varieties (Dunn’s Kruskal–Wallis multiple comparison, *p* < 0.05). Among these, TN1 (ditto; 71.1%) was most susceptible, followed by Beu Sim (ditto; 69.1%). The other three varieties, Hawm Dawk Doo, KDML105, and Gled Plah Chawn were categorized as intermediate (I) and did not significantly differ in the AUC from resistant or susceptible varieties. No varieties were categorized as highly susceptible (HS).

### 3.2. Scanning Electron Microscopy (SEM) Observation

SEM observed the physical morphology of rice trichomes in 10 different varieties, revealing two distinct types: (1) glandular trichomes (Figure 1C arrowhead), which were found in all rice varieties; and (2) non-glandular trichomes. The non-glandular trichomes were classified into four types as follows: (2.1) prickle trichomes (Figure 1A arrowhead) from RH, Suphan Buri 1, Hawm Dawk Doo, Gled Plah Chawn, Kam Pai, TN1, and Beu Sim; (2.2) macro trichomes (Figure 1B arrowhead) from all 10 selected varieties; (2.3) micro trichomes (Figure 1C arrowhead) from RH, Suphan Buri 1, Gled Plah Chawn, Sahm Ruang and Beu Sim; and (2.4) Papillae (Figure 1D arrowhead).

The length of trichomes observed from 10 selected rice varieties was measured to clarify the possibility of physical resistance (Table 3). The longest glandular trichome, prickle, and papillae were observed in Hom Dawk Doo. The longest micro and macro trichomes were observed from Gled Plah Chawn and Khao Maew, respectively. Khao Maew showed the highest density of all types of trichomes and those of glandular, macro, and papillae trichomes. Densities in RH were not as high compared with other varieties, with the exception of the prickle trichome, which showed the second–highest density among the 10 varieties (Table 4). According to the results of Spearman’s rank correlations (Table 5), the injury levels exhibited negative correlations in length to micro trichome, macro trichome, and papillae, although the correlations were not significant (*p* > 0.05). Negative correlations with the injury level were also recognized in the density of all trichomes and those of prickle trichomes, micro trichomes, and papillae. Among these, only the density of the prickle trichome showed a significantly negative correlation (*p* < 0.05). Macro trichome also showed a significant, but positive, correlation.

### 3.3. Preference of Brown Planthopper to VOC Blends from Different Rice Plants

According to the two–choice tests using uninfested rice plants, only Gled Plah Chawn was more preferred than RH, but not significantly (Figure 2) (*p* > 0.05, logistic regression). All the other varieties were less preferred than RH, and significant effects were recognized in the varieties KDML105, Suphan Buri 1, and TN1 (*p* < 0.05, logistic regression).

When using rice plants infested by BPH (Figure 3), the number of varieties that were less preferred than RH decreased from eight to four. Among these, only Suphan Buri 1 and TN1 were significant (*p* < 0.05, logistic regression). However, those more preferred increased from one to five, although significant effects were found only in Hawm Dawk Doo and Gled Plah Chawn (*p* < 0.05, logistic regression). Compared with the results before BPH infestation, the relative preference for RH, which was evaluated by the coefficients, generally increased in all the other varieties with smaller coefficient values than before BPH infestation.

### 3.4. VOCs 10 Rice Varieties before and during BPH Infestation

A total of 45 VOCs were identified from rice plants without BPH infestation. The greatest numbers of compounds were identified in the RH variety (17 compounds), followed by Gled Plah Chawn (12) and TN1 (8) (Table 6). On the other hand, the number was smallest in Hawm Dawk Doo and Beu Sim (2 compounds in each) followed by Sahm Ruang and KDML105 (3 in each). Among these, compounds that were recorded as insect repellents were included, such as Naphthalene and 1,2,3,4,4a,7-hexahydro-1,6-dimethyl-4-(1-methylethyl). The RH also contained the highest variety of sesquiterpenes (α-Cubebene, (Naphthalene, 1,2,3,4,4a,7-hexahydro-1,6-dimethyl-4-(1-methylethyl)-), and β-Curcumene).

A total of 18 VOCs were identified from rice plants with BPH infestation. Of these, 9 among the 18 were not detected from those without BPH. On the other hand, 36 VOCs were detected from rice plants without BPH infestation but not from those with BPH. The VOC diversity decreased greatly when the rice plants were infested with BPH. The greatest numbers of compounds were identified in the Gled Plah Chawn and Khao Maew varieties (5 compounds in each), while no compounds were detected from Hawm Dowk Doo and Kam Pai. The β-Sesquiphellandrene was found only in the RH variety with BPH. Naphthalene was found from Suphan Buri 1, Khao Maew, and TN1, both with and without BPH but not from RH without BPH. 

## 4. Discussion

The results of the evaluation of resistance on the chosen rice varieties conformed to previous studies that found that RH and Suphan Buri 1 were resistant to BPH infestation and had broad-spectrum resistance [1,45,46]. RH, a Sri Lanka landrace, is one of the most substantial sources of durable BPH resistance [47]. BPH3, BPH32, and the sesquiterpene synthase II (OsSTPS2) were identified as BPH resistance loci in RH [48,49,50]. RH has been used as a donor in Thailand’s and Southeast Asia’s breeding programs to improve BPH resistance performance [30]. Suphan Buri 1 is a high-yielding Thai rice variety resistant to BPH [32]. KDML105 and TN1 are susceptible to all BPH biotypes [31,51,52]; therefore, in this study, they were considered standard resistant and susceptible varieties compared with selected indigenous varieties. Injury levels by BPH were determined by the AUC calculated using scores from three timepoints (7, 9, and 11 DAI) according to the SSBS method by IRRI. Injury level was lowest in RH, followed by Suphan Buri 1 among the four standard varieties, and highest in TN1 (Table 2), which did not differ from the previous knowledge. Then the resistance of each variety was evaluated by the relative value of the AUC. The results of the four standard varieties were consistent with the previous reports, with the exception that KDML105 was evaluated as I. The other indigenous varieties, other than Hawm Dawk Doo, received higher injury than KDML105 and were evaluated as I or S.

Trichomes on plants are generally considered a physical defense in various plant species [53,54]. The morphology and density of trichome on rice plants are a subject of focus [55]. Although a significant negative relationship was found between the density of the prickle trichome and the injury level, the density of the prickle trichome was less than 10% of that of all trichomes. The density of all trichomes and those of the other types of trichomes had no significant relationships with the injury level. This indicated that non-glandular and glandular trichomes were unlikely involved in physical defense as a barrier against BPH. According to a previous study, the PTB33 rice variety, was considered resistant to BPH infestation, but in the two–choice test experiment, the BPH chose to settle on PTB33 rather than TN1, the most susceptible species in this study (Table 2) [56]. The wild rice (IRGC104646) showed longer trichomes, and PTB33 possessed higher trichome density, but they were less resistant to BPH than a variety (IRGC99577) with shorter trichomes and lower trichome density [56]. Several studies have been published on trichomes that have the ability to resist insects. Tomatoes with higher densities of glandular trichomes can have higher resistance (repellence) to spider mites [57]. Densely pubescent soybean has the potential to resist bean leaf beetle feeding on pods [27]. Furthermore, the most prevalent source of entrapment was unique trichomes, such as hooked trichomes in some plants, followed by puncturing or feeding, and, in rare circumstances, walking or fighting each body part of the insect [53]. In our study, the physical defense by trichomes was not effective against BPH infestation probably because trichomes were not tough enough to prevent colonization by second and third instars of BPH.

However, glandular trichomes have been known as epidermal outgrowths capable of biosynthesis and storage of large quantities of specialized metabolites that have a role in self–defense against biotic and abiotic stresses [58]. Recent studies identified plant VOCs in both constitutive and induced defense. Various terpene and derivatives were collected from the constitutive defense of selected rice varieties, for example, α-Cubebene, β-Curcumene, D-Limonene, β-Myrcene, Naphthalene, and so on. These terpenes have been recognized as insect repellents and have been used in many anti-insect products. Interestingly, regarding the preference of BPH for the volatile from rice plants without BPH infestation, BPH preferred RH, which is the most resistant variety (RH), rather than another resistant variety (Suphan Buri 1) and even on susceptible varieties such as KDML105 and TN1 (Appendix A). The VOC profile of intact rice plants was most diverse in RH, with 18 compounds among the 10 varieties. It can be presumed from these facts that BPHs were attracted by those diverse terpenes and derivatives emitted from RH. Therefore, our study found that RH may have no antixenosis mechanism of resistance, and BPH preferred volatiles emitted from the intact RH variety. Even with the induced defense in RH, BPHs still chose RH compared with Suphan Buri 1 and TN1. However, relative preference to RH decreased during BPH infestation in the other nine varieties. Rice aroma compounds probably affected attraction to BPH preference [59]. For example, β-Caryophyllene, which is a constitutively produced compound, is used to locate and recognize by BPH. According to our results, Gled Plah Chawn emitted Caryophyllene during BPH infestation, which agreed with the choice–test that the number of BPH taken on the Gled Plah Chawn side was greater than RH [60].

Previous studies on resistance mechanisms indicated that RH had second metabolites that affected BPH feeding and food foraging behavior. Those studies revealed that RH had antixenosis and antibiosis mechanism of resistance to BPH. These characteristics are relevant to the Sesquiterpene Synthase gene that performs a common role in response to herbivore attacks on rice plants [25,61]. Interestingly, RH was reported to release the major sesquiterpenes induced by BPH as follows: β-ionone, β-ionone epoxide, E-β-farnesene, and linalool within 10 days of BPH infestation [61]. Our experiments did not find those compounds in the volatile collecting chamber within 3 h of BPH infestation. This evidence can indicate that antixenosis in RH is not a quick response to the stresses, but it needs time or ambient mediated compounds of interaction among the rice plants. However, considering further the comparison of RH vs. Hom Dawk Doo and RH vs. Gled Plah Chawn, the number of BPH that chose RH was less than those indigenous varieties. Naphthalene was not found in RH, Hom Dawk Doo but was in Gled Plah Chawn, indicating that Naphthalene may perform an essential role as a quick responsive compound suddenly expressed after BPH infestation. Our results suggested that the diverse terpenes and derivatives are constitutive of the resistant mechanism in rice plants. They can be precursor substances to produce other compounds in response to BPH infestation. However, our findings showed that trichomes were not involved with physical defense. On the other hand, glandular trichomes perform a crucial role by being the location where the rice plant emits VOCs and large quantities of secondary metabolites to interact with abiotic and biotic stresses in rice plants.

## 5. Conclusions

The physical and chemical defense of resistant varieties (RH and Suphan Buri 1) and susceptible varieties (KDML105 and TN1) were compared to those of indigenous varieties (Hawm Dawk Doo, Gled Plah Chawn, Sahm Ruang, Kam Pai, Khao Maew, and Beu Sim). Resistance/susceptibility of indigenous varieties was evaluated as intermediate or susceptible. The length and density of leaf trichome unlikely prevented BPH infestation. However, glandular trichomes contained secondary metabolites interacting with abiotic and biotic stresses in rice plants. The most diverse terpene and derivatives were found in volatiles in intact rice plants of the resistant variety (RH), which can be the precursor compounds that responded to BPH. The VOC profiles changed greatly after the BPH infestation started. Relative preference to the RH by BPH decreased when the BPH infestation started. The newly emerged sesquiterpenoids, β-Sesquiphellandrene, may be involved in the result. This study identified Naphthalene as a dominant VOC inducible emitted during 3 h against BPH infestation. This finding may open a path to understanding how rice plants instantly respond to BPH infestation.

## Figures and Tables

**Figure 1 insects-13-00427-f001:**
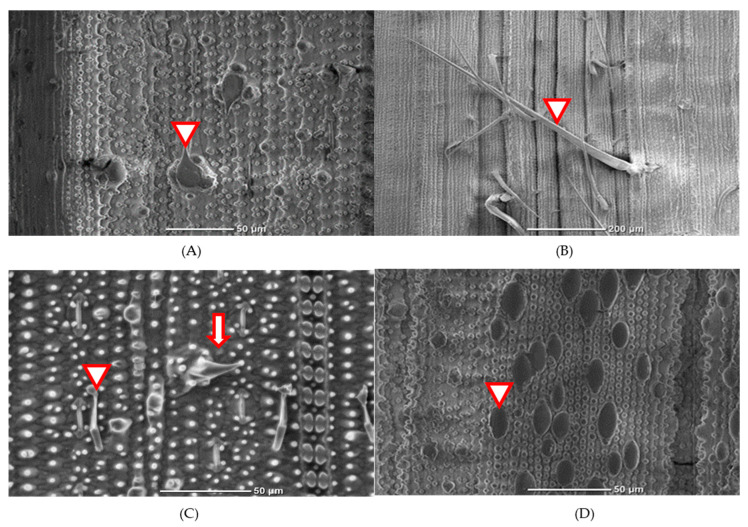
The scanning electron microscopy (SEM) observation of rice leaf surface at 500× magnification. Rice trichomes found in selected varieties were classified as follows (**A**) arrowhead: prickle trichome; (**B**) arrowhead: macro trichome; (**C**) arrow: micro trichome; arrowhead: glandular trichome; and (**D**) arrowhead: Papillae.

**Figure 2 insects-13-00427-f002:**
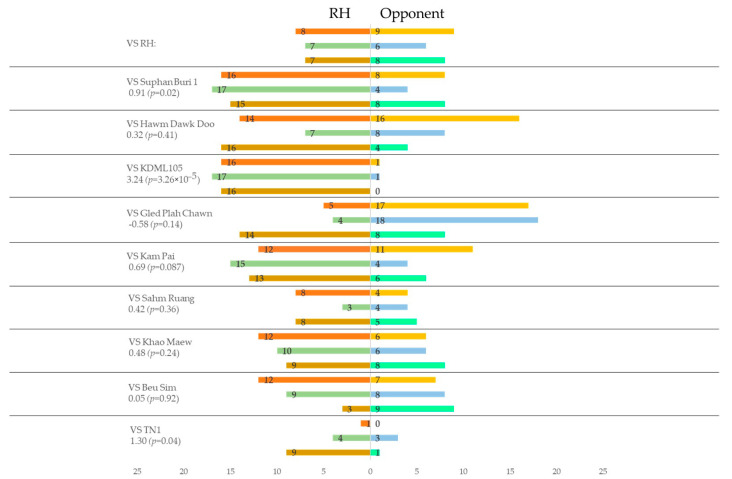
Relative olfactory preferences of BPH for volatiles from rice plants of RH variety to the others determined by two−choice tests using potted plants without BPH infestation.

**Figure 3 insects-13-00427-f003:**
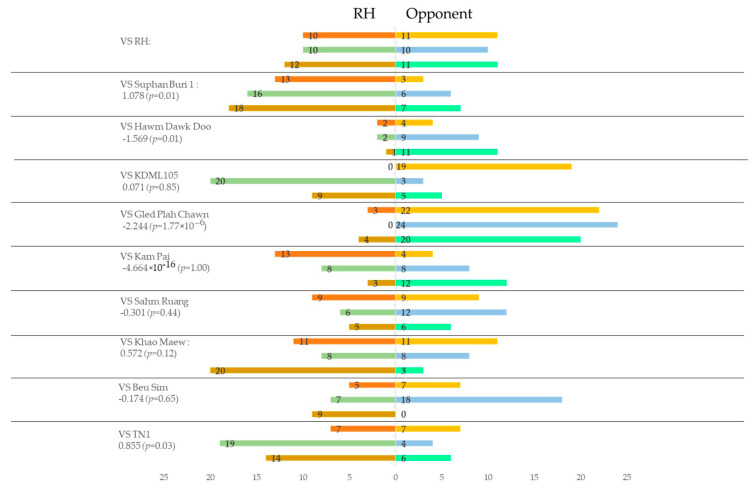
Relative olfactory preferences of BPH for volatiles from rice plants of RH variety to the others determined by two–choice tests using potted plants with BPH infestation.

**Table 1 insects-13-00427-t001:** Ten rice varieties used in the experiments.

Variety	Varieties Description	Originated	Resistance to BPH	Reference
Beu Sim	Indigenous rice variety	Thailand	Data not found	-
Gled Plah Chawn	Indigenous rice variety	Thailand	Data not found	-
Hawm Dawk Doo	Indigenous rice variety	Thailand	Data not found	-
Kam Pai	Indigenous rice variety	Thailand	Data not found	-
Khao Maew	Indigenous rice variety	Thailand	Data not found	-
Sahm Ruang	Indigenous rice variety	Thailand	Data not found	-
Rathu Heenati (RH)	Resistant donor variety in the conventional breeding program	Sri Lanka	Resistant	[30]
Taichung Native 1 (TN1)	Standard susceptible variety	Taiwan	Susceptible	[31]
Khao Dawk Mali 105 (KDML105)	Commercial variety	Thailand	Susceptible	[32]
Suphan Buri 1	Commercial variety	Thailand	Resistant	[32]

**Table 2 insects-13-00427-t002:** Injury levels by BPH obtained by SSBS test and evaluated BPH resistance of 10 rice varieties.

Varieties	Injury Score by SES (0–9)			BPH Damage AUC	Evaluated BPH Resistance
	7 DAI	9 DAI	11 DAI		
RH	0.0 (0–0) ^a^	0.2 (0–1) ^a^	6.0 (3–9) ^ab^	3.2 (1.5–4.5) ^a^	HR
Suphan Buri 1	0.0 (0–0) ^ab^	2.0 (1–3) ^ac^	6.7 (5–9) ^a^	5.3 (3.5–7.5) ^ac^	R
Hawm Dawk Doo	0.3 (0–1) ^ab^	4.2 (0–7) ^abc^	9.0 (9–9) ^c^	8.8 (4.5–11.5) ^abc^	I
KDML105	0.7 (0–3) ^ab^	5.3 (5–7) ^bc^	8.7 (7–9) ^c^	10.0 (8.5–12.0) ^bc^	I
Gled Plah Chawn	1.3 (0–5) ^ab^	5.0 (3–7) ^bc^	9.0 (9–9) ^c^	10.1 (7.5–14.0) ^abc^	I
Kam Pai	2.3 (0–5) ^ab^	5.0 (1–7) ^bc^	9.0 (9–9) ^c^	10.6 (5.5–14.0) ^b^	S
Sahm Ruang	2.2 (0–5) ^ab^	5.3 (5–7) ^bc^	8.3 (5–9) ^bc^	10.6 (8.0–14.0) ^b^	S
Khao Maew	3.2 (0–7) ^ab^	5.3 (3–7) ^bc^	9.0 (9–9) ^c^	11.4 (7.5–15.0) ^b^	S
Beu Sim	1.8 (0–3) ^ab^	6.3 (5–7) ^b^	9.0 (9–9) ^c^	11.8 (9.5–13.0) ^b^	S
TN1	3.8 (0–7) ^b^	5.7 (1–9) ^b^	9.0 (9–9) ^c^	12.1 (5.5–17.0) ^b^	S
Mean	1.0	4.6	8.5	9.4	
*p*-value	0.006	0.0007	0.0002	0.0004	

Mean (minimum–maximum) are shown. DAI: days after infestation. Same letter indicates no statistical difference at *p* = 0.05. (Post Hoc Dunn’s multiple comparison using adjusted *p*-values by the Benjamini–Hochberg method following Kruskal–Wallis rank-sum test). Evaluated resistance to BPH was categorized using the relative value of the AUC. HR: highly resistant; R: resistant; I: intermediate; S: susceptible; Criteria for the categories are shown in the text.

**Table 3 insects-13-00427-t003:** Length of trichomes on the leaf surface of 10 rice varieties.

Rice Varieties	Glandular Trichome	Non-Glandular Trichome			
		Prickle	Micro	Macro	Papillae
RH	35.10 ± 0.49 ^b^	36.00 ± 1.80 ^d^	16.80 ± 0.75 ^c^	247.67 ± 12.44 ^b^	15.80 ± 0.57 ^bc^
Suphan Buri 1	37.93 ± 3.66 ^ab^	42.77 ± 3.84 ^cd^	16.67 ± 2.50 ^c^	170.33 ± 17.46 ^c^	15.60 ± 0.92 ^bc^
Hom Dawk Doo	44.83 ± 2.23 ^a^	74.47 ± 4.37 ^a^	0.00 ± 0.00 ^d^	249.33 ± 13.78 ^b^	19.00 ± 0.25 ^a^
KDML105	38.03 ± 1.42 ^ab^	0.00 ± 0.00 ^e^	0.00 ± 0.00 ^d^	157.33 ± 20.85 ^cd^	18.07 ± 1.01 ^ab^
Gled Plah Chawn	40.23 ± 4.36 ^ab^	55.10 ± 7.95 ^b^	25.53 ± 1.58 ^a^	125.33 ± 0.88 ^cd^	14.50 ± 0.75 ^c^
Kam Pai	42.37 ± 0.27 ^ab^	45.83 ± 1.67 ^bcd^	21.40 ± 2.63 ^b^	110.80 ± 5.88 ^d^	14.83 ± 2.09 ^c^
Sahm Ruang	40.23 ± 2.14 ^ab^	0.00 ± 0.00 ^e^	23.03 ± 0.22 ^ab^	229.33 ± 7.31 ^b^	15.33 ± 0.60 ^bc^
Khao Maew	40.20 ± 0.87 ^ab^	0.00 ± 0.00 ^e^	0.00 ± 0.00 ^d^	342.67 ± 34.23 ^a^	15.00 ± 0.31 ^c^
Beu Sim	40.87 ± 1.91 ^ab^	48.47 ± 4.60 ^bc^	22.73 ± 0.22 ^ab^	162.33 ± 9.96 ^cd^	16.00 ± 0.26 ^bc^
TN1	37.90 ± 1.65 ^ab^	43.23 ± 4.43 ^cd^	0.00 ± 0.00 ^d^	108.13 ± 19.45 ^d^	14.53 ± 0.52 ^c^

Mean ± SD. The same superscript letters indicate no statistical difference at *p* = 0.05 (Post Hoc Duncan’s multiple range test following ANOVA).

**Table 4 insects-13-00427-t004:** Density of trichomes on the leaf surface of 10 rice varieties.

Rice Varieties	All Trichomes	Glandular Trichome	Non-glandular Trichome			
			Prickle	Micro	Macro	Papillae
RH	11.42 ± 0.58 ^c^	0.33 ± 0.38 ^e^	0.58 ± 0.14 ^b^	0.42 ± 0.14 ^bc^	0.17 ± 0.14 ^e^	9.92 ± 0.52 ^c^
Suphan Buri 1	10.92 ± 1.04 ^cd^	1.75 ± 0.43 ^c^	1.58 ± 1.26 ^a^	0.83 ± 0.29 ^a^	0.42 ± 0.14 ^de^	6.33 ± 0.88 ^cde^
Hom Dawk Doo	15.67 ± 3.25 ^b^	2.50 ± 0.50 ^b^	0.17 ± 0.14 ^b^	0.00 ± 0.00 ^d^	2.58 ± 0.95 ^a^	10.42 ± 3.75 ^b^
KDML105	8.42 ± 0.58 ^cde^	0.67 ± 0.14 ^de^	0.00 ± 0.00 ^b^	0.00 ± 0.00 ^d^	1.75 ± 0.25 ^b^	6.00 ± 0.50 ^de^
Gled Plah Chawn	5.42 ± 1.28 ^e^	0.83 ± 0.14 ^de^	0.17 ± 0.14 ^b^	0.67 ± 0.29 ^ab^	0.17 ± 0.29 ^e^	3.58 ± 0.95 ^e^
Kam Pai	7.75 ± 1.98 ^de^	0.58 ± 0.14 ^de^	0.17 ± 0.29 ^b^	0.92 ± 0.38 ^a^	0.92 ± 0.29 ^cde^	5.17 ± 2.57 ^de^
Sahm Ruang	11.50 ± 1.64 ^c^	2.58 ± 0.63 ^b^	0.00 ± 0.00 ^b^	0.17 ± 0.14 ^cd^	0.67 ± 0.29 ^cde^	8.08 ± 1.04 ^bcd^
Khao Maew	21.33 ± 2.01 ^a^	3.67 ± 0.38 ^a^	0.00 ± 0.00 ^b^	0.00 ± 0.00 ^d^	2.50 ± 0.25 ^a^	15.17 ± 2.27 ^a^
Beu Sim	8.25 ± 1.15 ^cde^	1.25 ± 0.43 ^cd^	0.25 ± 0.00 ^b^	0.25 ± 0.25 ^cd^	1.08 ± 0.52 ^bcd^	5.42 ± 1.59 ^de^
TN1	9.33 ± 3.26 ^cd^	1.75 ± 0.66 ^c^	0.08 ± 0.14 ^b^	0.00 ± 0.00 ^d^	1.33 ± 0.29 ^bc^	6.17 ± 3.06 ^de^

Densities of trichomes per square millimeters (mean ± SD) are shown. The same superscript letters indicate no statistical difference at *p* = 0.05 (Post Hoc Duncan’s multiple range test following ANOVA).

**Table 5 insects-13-00427-t005:** Spearman’s rank correlation (ρ) of injury level and the physical characteristics of trichomes on 10 rice varieties.

	Length	Density
All trichomes	NA	−0.15 (*p* = 0.42)
Grandular trichome	0.05 (*p* = 0.80)	0.27 (*p* = 0.15)
Prickle trichome	0.05 (*p* = 0.79)	−0.43 (*p* = 0.016)
Micro trichome	−0.10 (*p* = 0.60)	−0.33 (*p* = 0.08)
Macro trichome	−0.32 (*p* = 0.08)	0.44 (*p* = 0.016)
Papillae	−0.10 (*p* = 0.60)	−0.20 (*p* = 0.27)

NA: Not Available.

**Table 6 insects-13-00427-t006:** Compounds identified in volatiles from 10 varieties of rice plants with (w) and without (w/o) BPH infestation.

	Compounds	RH		Suphan Buri 1		Hawm Dawk Doo		KDML105		Gled Plah Chawn		Kam Pai		Sahm Ruang		Khao Maew		Beu Sim		TN1	
		w/o	w	w/o	w	w/o	w	w/o	w	w/o	w	w/o	w	w/o	w	w/o	w	w/o	w	w/o	w
Monoterpenoids	Total	0	0	2	0	0	0	0	0	1	1	0	0	0	0	0	0	0	0	2	0
	Bicyclo[3.1.0]hex-2-ene,4-methyl-1-(1-methylethyl)-																			√	
	D-Limonene			√																	
	α-Phellandrene			√																√	
	β-Myrcene									√	√										
Sesquiterpenoids	Total	2	1	1	0	0	0	0	0	0	1	1	0	0	1	1	1	0	0	1	0
	(S,1Z,6Z)-8-Isopropyl-1-methyl-5-methylenecyclodeca-1,6-diene																√				
	Aromandendrene											√									
	Caryophyllene										√					√					
	α-Cubebene	√		√											√						
	β-Caryophyllen																			√	
	β-Curcumene	√																			
	β-Sesquiphellandrene		√																		
Norsesquiterpenes	Total	0	0	0	0	0	0	0	0	0	1	0	0	0	0	0	0	0	0	0	0
	Geijerene										√										
Carbonyl compounds	Total	3	0	0	0	0	0	0	0	0	0	0	0	0	0	0	1	0	0	0	2
	1-Hepten-3-one	√																			
	1-Penten-3-one																				√
	4′-Ethylpropiophenone	√															√				
	Ethanone,1-(4-ethylphenyl)-	√																			√
Terpene	Total	12	2	3	1	2	0	3	1	11	2	5	0	3	2	3	3	2	3	5	2
	1-Methyl-3-(1′-methylcyclopropyl)cyclopentene									√											
	1-Octene,2-methyl-	√																			
	1,3-Bis(cyclopentyl)-1-cyclopentanone																			√	
	1,3-Cyclohexadiene,5,6-dimethyl-									√											
	1,3,4,6-Hexanetetrone,1-(4-methylphenyl)-6-phenyl-									√											
	1,5-Heptadiene,(E)-																				√
	1,5-Heptadiene,(Z)-											√									
	1,5-Heptadiene,3-methyl-,(E)-	√																			
	1H-Indene,1-hexadecyl-2,3-dihydro-																			√	
	2,5-Cyclohexadien-1-one,4,4′-(1,2-ethanediylidene)bis[2,6-bis(1,1-dimethylethyl)-														√						
	3-Ethyl-3-hexene	√																			
	3-Pentanone,2,2,4,4-tetramethyl-									√											
	3-Undecene,5-methyl-	√																			
	4-(2′,4′,4′-trimethyl-yciclo[4.1.0]hept-2′-en-3′-yl)-3-buten-2-one																	√			
	4-Undecene,6-methyl-	√																			
	5-Hepten-3-one,5-ethyl-4-methyl-	√		√																	
	Benzene,(2,2-dimethylpropyl)-					√															
	Benzene,[(cyclohex-1-en-1-yl)methyl]-					√															
	Benzene,1,2-diethyl-													√							
	Benzene,1,3-diethyl																		√		
	Benzene,1,4-diethyl-													√		√	√		√		
	Bicyclo[5.2.0]nonane,2-methylene-4,8,8-trimethyl-4-vinyl-									√											
	cis-Muurola-4(15),5-diene		√					√													
	Cyclobutane,1,2-bis(1-methylethenyl)-,trans-																			√	
	Cyclobutane,1,2-dipropenyl-																		√		
	Cyclobutane,1,3-diisopropenyl-,trans	√	√					√		√	√	√				√	√	√			
	Cyclobutanone,2,3,3,4-tetramethyl-														√						
	Cyclohexene,1-(1-propynyl)-									√											
	Cyclopentanone,2,2,5-trimethyl-													√							
	Cyclopropane,1-(2-methylbutyl)-1-(1-methylpropyl)-	√																			
	Decane,3,8-dimethyl-	√																			
	Hexane,3-methyl-4-methylene-	√																			
	Hexane,3,4-bis(1,1-dimethylethyl)-2,2,5,5-tetramethyl-			√						√		√								√	
	Naphthalene			√	√			√	√	√	√					√	√			√	√
	Naphthalene,1,2,3,4,4a,7-hexahydro-1,6-dimethyl-4-(1-methylethyl)-	√								√											
	Octadecane,2,2,4,15,17,17-hexamethyl-7,12-bis(3,5,5-trimethylhexyl)-											√									
	Santolinatriene									√											
	Tridecane,2,2,4,10,12,12-hexamethyl-7-(3,5,5-trimethylhexyl)-	√										√									
Sum total		17	3	6	1	2	0	3	1	12	5	6	0	3	3	4	5	2	3	8	4

## Data Availability

The data presented in this study are available on request from the corresponding author.

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
