# Peer review of "Mechanisms of Trichomes and Terpene Compounds in Indigenous and Commercial Thai Rice Varieties against Brown Planthopper"

_insects, 2022, doi:10.3390/insects13050427_

Round 1

Reviewer 1 Report

The manuscript entitled "Mechanisms of Trichomes and Terpenes compound in Indigenous and Commercial Thai Rice Varieties  Against Brown Planthopper" has shown the research about the trichomes in different rice varieties and chemical evidence of  induced resistance during rice interacted with BPH. The work sounds interesting, but this manucript has many logical and  careless mistakes, lack of introduction and explanation of method, overstated conclusion. Becides, this manuscript needs more  attention to English grammar, spelling and sentence structure so that the goals and results of the study would be clear to  readers. The following are the questions and some mistakes in this manuscript:
1. Page 1 Line 22, the authors stated "morphological structures known as trichomes or antibiosis and antixenosis traits" antibiosis and antixenosis could elucidate the interaction between plants and insects. But trichomes are morphological  structure that might function in antibiosis or/and antixenosis. The authors stated them together, that would make readers  confused. 
2. Page1 Line 27, the authors stated "dense trichomes are infective in terms of preventing BPH infestation". According to the  study, trichome density are useless in BPH resistance. Thus, the "infective" should be useless or ineffective. And "in terms  of"could be revised to "in", so that the statement would be clear and simple.
3. Page 1 Line 40-42, the authors noted "When BPHs were fed rice, essential oils that primarily impacted herbivores confined  in the most resistant rice variety (RH) ceased to emit." The sentence structure seems so confused and it makes reader hard to understand. Subordinate clause should be organized correctly. Such problems could be found throughout the whole text, 
including Page 2 Line 52-53, and so on.
4. Some conclusions are overstated. For example, in many cases, trichome is responsible for insect feeding, oviposition  responses,and the nutrition of larvae. Specialized hooked trichomes may impale adults or larvae as well. The authors only  recorded the length of trichomes, to conclude the injury level is not related with the physical characteristics of rice  trichomes. Even authors stated "The density and length of trichome on each variety were ineffective in preventing BPH".
Page 17 Line 424-425 "glandular trichome is crucial site of biosynthesis and storage of large quantities of secondary  metabolites" the authors only tested VOCs from rice, why they could say "biosynthesis and storage of large quantities of  secondary metabolites"?
5. For Page 3 "the percentage of injury levels", the calculation method seems not rational. No any reference to make it persuasive. The method was based on the straight-line graph, but no articles show the infestation is linear with scores.  Authors need to give more explanation. As for the Table 3, authors need explanation to show why 0-20 was defined as highly resistant. If the injury level is 40.05, would the authors consider it as resistant or moderate?
6. "RH" referred to Rathu Heenati, but the authors did not mention it clearly in the text. Good article should be clear to readers.
7.For all the data, the authors did not show how many replicates or sample number they were used.
8. Some spelling error in the text, such as "Antixenis" Page 17 Line 436,  "towas" Page 17 Line 438 etc. Please pay more attetion to the manuscript.
9.Even the authors made some hypothesis, the study design or the results can not support their ideas correctly. The unexpeted results were not stated and explained enough. Therefore, the part of conclusion seems overstated. For example, in "3.3 VOC Identification of Rice Terpenes and Derivatives (Constitutive and Induced defenses) and behavioral responses of Brown planthopper toward different rice-sources",the author did not use any rice variety that has been reported with VOCs to make your method effective first.

10. Figure 5 is just listing the origin results, without any analysis. If authors could devide the chemicals into several groups, it would make readers more clear to understand.

Author Response

The manuscript entitled "Mechanisms of Trichomes and Terpenes compound in Indigenous and Commercial Thai Rice Varieties  Against Brown Planthopper" has shown the research about the trichomes in different rice varieties and chemical evidence of  induced resistance during rice interacted with BPH. The work sounds interesting, but this manucript has many logical and  careless mistakes, lack of introduction and explanation of method, overstated conclusion. Becides, this manuscript needs more attention to English grammar, spelling and sentence structure so that the goals and results of the study would be clear to  readers. The following are the questions and some mistakes in this manuscript:

  • First, we would like to apologize for being negligent and making mistakes that confused the reader. We proved the grammar and syntax throughout the manuscript. We have already corrected the sentences as you recommend.
  1. Page 1 Line 22, the authors stated "morphological structures known as trichomes or antibiosis and antixenosis traits" antibiosis and antixenosis could elucidate the interaction between plants and insects. But trichomes are morphological  structure that might function in antibiosis or/and antixenosis. The authors stated them together, that would make readers  confused. 
  • We have edited and added the sentences to

“Plants produce specialized morphological structures known as trichomes, which may be involved in the antibiosis and antixenosis traits of the host plant when interacting with insect herbivores. The host plant also produces chemicals that may act as repellent or antifeedant and reduce herbivore performance.” (Line 27-30).

  1. Page1 Line 27, the authors stated "dense trichomes are infective in terms of preventing BPH infestation". According to the  study, trichome density are useless in BPH resistance. Thus, the "infective" should be useless or ineffective. And "in terms of"could be revised to "in", so that the statement would be clear and simple.
  • We have edited to “In this study, we found a negative relationship between the density of prickle trichomes the BPH infestation level.” (Line 33)
  1. Page 1 Line 40-42, the authors noted "When BPHs were fed rice, essential oils that primarily impacted herbivores confined  in the most resistant rice variety (RH) ceased to emit." The sentence structure seems so confused and it makes reader hard to understand. Subordinate clause should be organized correctly. Such problems could be found throughout the whole text, 
    including Page 2 Line 52-53, and so on.

  • We have edited to “When rice plants were infested by BPH, the numbers of VOCs and these quantities decreased.” (Line 46-47)
  1. Some conclusions are overstated. For example, in many cases, trichome is responsible for insect feeding, oviposition responses, and the nutrition of larvae. Specialized hooked trichomes may impale adults or larvae as well. The authors only recorded the length of trichomes, to conclude the injury level is not related with the physical characteristics of rice trichomes. Even authors stated "The density and length of trichome on each variety were ineffective in preventing BPH".

  • Page 17 Line 424-425 "glandular trichome is crucial site of biosynthesis and storage of large quantities of secondary  metabolites" the authors only tested VOCs from rice, why they could say "biosynthesis and storage of large quantities of  secondary metabolites"?
  • Probably the reviewer would like to ask about Page 17 . It came from literature cited.
    For Page 3 "the percentage of injury levels", the calculation method seems not rational. No any reference to make it persuasive. The method was based on the straight-line graph, but no articles show the infestation is linear with scores.  Authors need to give more explanation. As for the Table 3, authors need explanation to show why 0-20 was defined as highly resistant. If the injury level is 40.05, would the authors consider it as resistant or moderate?
  • We used statistics to reanalyze the data and revised it in Line: 214-228. The result presented in Table 2.

  1. "RH" referred to Rathu Heenati, but the authors did not mention it clearly in the text. Good article should be clear to readers.
  • We checked and replaced throughout manuscript as the reviewer suggested.
    For all the data, the authors did not show how many replicates or sample number they were used.
  • We revised in Materials and Methods as the reviewer suggested.
  1. Some spelling error in the text, such as "Antixenis" Page 17 Line 436,  "towas" Page 17 Line 438 etc. Please pay more attetion to the manuscript.
  • We checked misspelling throughout manuscript as the reviewer suggested.

9.Even the authors made some hypothesis, the study design or the results can not support their ideas correctly. The unexpeted results were not stated and explained enough. Therefore, the part of conclusion seems overstated. For example, in "3.3 VOC Identification of Rice Terpenes and Derivatives (Constitutive and Induced defenses) and behavioral responses of Brown planthopper toward different rice-sources",the author did not use any rice variety that has been reported with VOCs to make your method effective first.

- We added “Terpenoid compounds of Rathu Heenati and their isogenic lines were identified by ex-traction at 750 C for 20 min before subjected to GC-MS from the previous study [39]. However, those VOCs were not trapped from spontaneous emission by from plants.” Line 83-85

  1. Figure 5 is just listing the origin results, without any analysis. If authors could devide the chemicals into several groups, it would make readers more clear to understand.

- The illustration of Figure 5 changed to Table 6 according to the third reviewer. Page 12 Line:350

Reviewer 2 Report

To the authors,

The manuscript titled ‘Mechanisms of Trichomes and Terpenes compound in Indigenous and Commercial Thai Rice Varieties Against Brown Planthopper’ is an interesting approach to decipher the mechanisms of resistance/susceptibility in rice varieties to BPH. The authors focused their study on trichome morphology-density and its correlation with volatile compound profiles and how these affect the insect host-preference behavior. However, the study needs improvements due to the following:

a) the times used as inducible and constitutive (3 hours) seem arbitrary, how the authors make the decision about this short time of VOC collection?;

b) observing the VOC profiles among varieties (Figure 5) gave me the impression that is not comparable despite they are rice varieties (plant breeding-parental selection, this information should be included), also a more extensive collection time or different collection times are needed to determine the host-preference of BPH to these varieties;

c) Statistics on tables should be denoted by superscript, as was shown in Table 7; and

d) Illustration of Figure 5 should be improved and also its experimental method description of the GC/MS analysis.

In addition to the previous paragraph, this manuscript needs English editing (small). Also, a few improvements might need it before publication, as was suggested. Finally, I believe this research has a lot of potential therefore I recommend a Major revision

Author Response

  1. a) the times used as inducible and constitutive (3 hours) seem arbitrary, how the authors make the decision about this short time of VOC collection?;
  2. observing the VOC profiles among varieties (Figure 5) gave me the impression that is not comparable despite they are rice varieties (plant breeding-parental selection, this information should be included), also a more extensive collection time or different collection times are needed to determine the host-preference of BPH to these varieties

    The VCTs were set up to trap volatile compounds for 3, 6, and 9 hrs. from 9:00 AM in preliminary test. There were no significant compounds trapped after 3 hrs., therefore only the results of 3 hrs. were used in our study. Statistical Analysis parts were reanalysis.

Reviewer 3 Report

This is  an interesting study focusing on the effect of trichome on injury levels caused by BPH, a significant pest of rice. The experiments are well designed, and the results are well presented. Therefore I think it can be accepted after some minor revisions. 

Table 1: Two cultivars, Rathu Heenati and Suphan Buri 1, have been studied for their resistance to BPH. These cultivars may provide valuable background information about resistance of rice against BPH. It would be better to provide detailed information and previous studies about these cultivars in Introduction.

Also, since many experiments were conducted, it would be better to provide a diagram to show the experimental design (e.g., the purpose of each experiment) and relationship among different experiments.  

Figure 5: This is a too large figure. I suggest presenting these data in a table or supplementary materials.

Table 7: The authors concluded that the physical traits of trichome did not significantly correlated with injury levels. However, I noticed that only a few parameters were measures (i.e., densities and length). Other factors, such as hardness and sharpness of trichome may also affect injury. Although it is not necessary to reconduct experiments at this time. it would be better to discuss these factors in Discussion.  

Author Response

Table 1: Two cultivars, Rathu Heenati and Suphan Buri 1, have been studied for their resistance to BPH. These cultivars may provide valuable background information about resistance of rice against BPH. It would be better to provide detailed information and previous studies about these cultivars in Introduction.

  • We added in to Discussion part in Line 372-375.

Also, since many experiments were conducted, it would be better to provide a diagram to show the experimental design (e.g., the purpose of each experiment) and relationship among different experiments.  

  • We revised in 2.1 Experimental design Line 105-125.

Figure 5: This is a too large figure. I suggest presenting these data in a table or supplementary materials.

  • We changed Figure 5 to Table 6. Page 12 Line:350

Table 7: The authors concluded that the physical traits of trichome did not significantly correlated with injury levels. However, I noticed that only a few parameters were measures (i.e., densities and length). Other factors, such as hardness and sharpness of trichome may also affect injury. Although it is not necessary to reconduct experiments at this time. it would be better to discuss these factors in Discussion.  

  • Thank you for your advised. We also added in Line:405-407

Round 2

Reviewer 1 Report

The revised manuscript entitled "Mechanisms of Trichomes and Terpenes compound in Indige-2 nous and Commercial Thai Rice Varieties Against Brown 3 Planthopper 4" has been improved much better than the original version. It provided more introductions and explanantion of the methods and data analysis. It is good for the audience of this journal, except several spelling mistakes as below:
1. Page 1 Line 34, "between the density of prickle trichomes the BPH infestation level" 'and' was missing.
2. Page 4 Line 147, the first AUC should be indicated with the whole name as"the average area under the curve"
3. Page 9 Line 323, "(the oefficient values decreased)" 'oefficient' should be coefficient, and the sentence in the brackets is easily misleading to readers. It could be expressed in a new sentence.

Author Response

Dear Reviewer 1.

Thank you for taking the time to leave your comment. We took the following steps in response to your comment: 

1. Page 1 Line 34, "between the density of prickle trichomes the BPH infestation level" 'and' was missing.

We done

2. Page 4 Line 147, the first AUC should be indicated with the whole name as"the average area under the curve"

We done
3. Page 9 Line 323, "(the oefficient values decreased)" 'oefficient' should be coefficient, and the sentence in the brackets is easily misleading to readers. It could be expressed in a new sentence.

We change as "Compared to the results before BPH infestation, the relative preference for RH, which was evaluated by the coefficients, generally increased in all the other varieties with smaller coefficient values than before BPH infestation." Line 343-345

We proved the grammar and syntax throughout the manuscript and the manuscript was edited by English language of MDPI. We have already corrected the sentences as you recommend.